# Propensity to childhood anxiety and depression due to exposure to adversity: A multidimensional construct

Angélica C. G. Santos[1], Ariane Silva[2], Matheus Libório[1], Cristiane Neri Nobre[2], Hasheem Mannan[3]*, Marcos Flávio S. V. D'Angelo[1]

**1** Postgraduate Program in Computational Modeling and Systems, Montes Claros State University, Montes Claros, Minas Gerais, Brazil, **2** Postgraduate Program in Informatics, Pontifical Catholic University of Minas Gerais, Belo Horizonte, Minas Gerais, Brazil, **3** University College Dublin School of Nursing, Midwifery and Health Systems, Health Sciences Centre, Belfield, Dublin, Ireland

* hasheem.mannan@ucd.ie

## Abstract

Children and adolescents are increasingly susceptible to issues related to anxiety and depression symptoms. The literature does not present a consensus on the composition of indicators that make predictions, prognostic algorithms, or management strategies in mental health promotion and prevention. Most studies primarily focus on the consequences observed in adulthood. This study develops a multidimensional representation of the propensity of children and adolescents to have difficulties in the field of anxiety and depression. The Ordered Weighted Averaging (OWA) operator was used to create a composite indicator, and three quality tests validated the results. For this, it uses information about different dimensions associated with adversity in childhood and adolescence from 54 countries sourced from UNICEF's Multiple Indicator Cluster Surveys to compare the values of proposed dimensions across continents. The generated composite indicator reveals that, on average, countries in Africa show a higher propensity for children and adolescents to present difficulties in the anxiety and depression fields. In the opposite position, the Americas have the lowest average propensity for these mental health conditions. The validation of the results through quality tests reinforces confidence in the direction indicated by the findings, enhancing the decision-making process when dealing with multidimensional phenomena.

## Introduction

Children and adolescents are increasingly susceptible to issues related to anxiety and depression symptoms. Approximately 6.5% of children and adolescents are estimated to suffer from anxiety, while 2.6% are affected by depression [1]. It is crucial to

**Data availability statement:** The one file is available from the Mendeley database. Gouveia dos Santos, Angélica Cidália (2024), "Data_Propensity to Childhood Anxiety and Depression Due to Exposure to Adversity", Mendeley Data, V1, doi: https://doi.org/10.17632/gzggks7t7p.1.

**Funding:** The author(s) received no specific funding for this work.

**Competing interests:** The authors have declared that no competing interests exist.

address mental health problems as they are becoming more prevalent and can lead to long-term consequences for individuals, families, and communities [2].

Mental health conditions, including depression, anxiety, and other emotional disorders, account for 15% of the global burden of diseases among adolescents aged 10–19 [3]. Early-life depression and anxiety threaten the future physical and mental health of children and adolescents, as well as their educational performance, financial behavior, and interpersonal relationships [4]. These implications hinder acquiring skills necessary for the transition to adulthood [5].

Measuring mental health conditions has been the subject of studies for developing various tools and scales, each with strengths and limitations [5]. Low-income and middle-low-income countries face difficulties obtaining key indicators and lacking consensus on mental health monitoring methods [5].

The causes or risks of mental disorders can be present at various levels, including genetic, neural, individual, family environment, and social context [6]. Mental health is a complex concept that involves multiple dimensions and various factors [7,8]. A multifaceted perspective provides a comprehensive understanding of mental health, although analyzing its various factors demands considerable cognitive effort [9,10].

Untreated mental health disorders in children and adolescents pose a significant public health burden, with stigmatization being a universal and debilitating issue [11]. More specifically, in cases associated with anxiety and depression disorder, adverse social and individual effects generated include higher healthcare costs, increased risk of physical comorbidities, and impacts on quality of life [12]. Symptoms of anxiety and depression often coexist, exacerbating the effects of many physical, cognitive, and psychosocial difficulties [5].

Depressive and anxiety disorders are associated with adverse childhood experiences such as child maltreatment (e.g., physical, emotional, sexual abuse, physical and emotional neglect) and family dysfunction (e.g., parental divorce or separation, incarceration, violence, mental illness, and substance abuse) [13–15]. Alarmingly, adverse childhood experiences are common, with about half of the United States population having experienced at least one adverse experience before adulthood [16]. This fact suggests that positive childhood experiences related to family, school, and neighborhood protect adolescents against depression and anxiety, while dysfunctions in these areas increase the risks of both [15,17].

Therefore, developing studies that highlight and evaluate the evidence on the factors that correlate with difficulties related to depression and anxiety in children and adolescents is crucial for the development of effective actions by public agents.

However, few studies are concerned with analyzing anxiety and depression in childhood from a multidimensional perspective [15]. There is no consensus on the composition of indicators that make predictions, prognostic algorithms, or management strategies in mental health promotion and prevention [18], and the focus is on the consequences observed in adult life [19].

Considering these gaps, this study develops a multidimensional representation of the propensity of children and adolescents to experience difficulties in the field of anxiety and depression. The childhood anxiety and depression composite indicator

is constructed using the Ordered Weighted Averaging (OWA) operator. This process incorporates information from three dimensions related to adversity experienced during childhood and adolescence.

This study contributes to mental health research by employing data from 54 countries sourced from UNICEF's Multiple Indicator Cluster Surveys to compare the values of proposed dimensions across continents. The study results also enhance understanding of the most pertinent dimensions of mental health in each continent. Finally, the study's findings contribute to ensuring the robustness of the composite indicator by evaluating its explanatory and discriminant power and the proportion of outlier measurements.

The Multiple Indicator Cluster Surveys (MICS) is an international household survey developed by UNICEF to fill data gaps and monitor human development. Since 1995, MICS has been conducted every five years, providing robust and internationally comparable statistical estimates of key indicators for monitoring global goals. In this study, we use data from the sixth round (MICS6), conducted between 2018 and 2019 by the Ministry of Economy and Finance. MICS6 covers 54 countries (this data was downloaded in March 2024) and includes 72 questionnaires, 177 core indicators, and an average sample of 12,000 households, making it the most extensive edition. The MICS questionnaires address demographic (age, ethnicity, gender), economic (income inequality, child labor), infrastructure, security, education, social support, and functional difficulties, allowing comparative analysis of the biopsychosocial conditions of children and adolescents globally.

The construction of the propensity to childhood anxiety and depression composite indicator proposed in this study explores gaps in methodologies for weighing sub-indicators associated with multidimensional anxiety and depression within the mental health field [20]. It also tests the quality of the composite indicator and its ability to represent the multidimensional phenomenon reliably.

The methodological approach proposed in this study also provides the means to differentiate countries based on the different dimensions of anxiety and depression. This differentiation helps decision-makers design targeted public policies, ensuring greater precision in the allocation of resources and efforts to prevent and assist childhood anxiety and depression. Furthermore, the study addresses the gap in analyses of factors negatively correlated with mental health in children and adolescents, particularly the growing incidence of anxiety and depression in this population [19] in low or lower-middle-income countries.

## Anxiety and depression and exposure to adverse childhood experiences

Contextualizing the factors that lead to anxiety and depression, numerous studies link the occurrence of anxiety and depression disorders to adverse childhood experiences [13,14,21]. These negative experiences are associated with potentially traumatic events occurring before the age of 18 [19,22] and encompass various forms of psychological, physical, and sexual abuse; neglect and domestic dysfunction; community violence; victimization; racial discrimination; and poverty [22].

Adverse childhood experiences are considered common. [21] indicate that individuals exposed to one or more types of adverse childhood experiences are 1.5 to 3.5 times more likely to experience depressed mood and 1.3 to 6.8 times more likely to report anxiety. In the United States, about half of the population has experienced at least one adverse experience before adulthood [16].

Research on adverse childhood experiences has predominantly focused on child maltreatment, particularly sexual and physical abuse, in high-income countries [23]. However, factors such as low socioeconomic status, discrimination, interparental conflict, parental mental and chronic illnesses, and exposure to violence or wars also elevate the risk of mental disorders in children [24]. The fact that each type of adverse childhood experience does not equally affect health limits score-based analyses, impacting the ability to conduct targeted prevention efforts [25].

From a public health perspective, the multidimensional and overlapping nature of adverse childhood experiences [26] necessitates collaboration across various sectors and stakeholders in different areas [24]. Interventions to prevent and mitigate the negative impacts of adverse childhood experiences should focus on promoting positive parenting behaviors,

preventing bullying and child abuse, improving socioeconomic status, and combating drug use, among other strategies [15,24].

Individuals exposed to four or more adverse childhood experiences have higher chances of experiencing anxiety, depression, and suicide attempts compared to those without adverse childhood experiences [23]. Given the high prevalence of adverse childhood experiences and the growing evidence of their contribution to most classes of mental disorders [16], adverse childhood experiences are strategic targets for prevention [24].

The relationship between adverse childhood experiences and the occurrence of anxiety and depression is widely addressed by literature [24]. However, few examine whether the type of adverse childhood experiences exposure differentially affects anxiety and depression outcomes among children and youth [19]. Furthermore, few studies are concerned with analyzing anxiety and depression from a multidimensional perspective and do not identify which factors associated with adverse childhood experiences are most relevant, making it difficult to define policies to prevent anxiety and depression [15].

## Multidimensionality of anxiety and depression

Mental health is influenced by many factors, including lifestyle choices [27], social and economic conditions, environmental influences [28], and individual psychological and biological characteristics [29]. In this context, various factors contribute to childhood anxiety and depression, including parental influences [30], childhood adversities [15], and genetic predispositions [31].

A practical solution to simplify the analysis and understanding of the various mental health factors is the construction of composite indicators [32,33]. This involves selecting and combining various individual, social, economic, and environmental variables to create a comprehensive measure of mental health at the population level [34].

Composite indicators combine multiple factors to provide a more thorough risk assessment than unidimensional approaches [35,36]. These characteristics are relevant for monitoring, promoting, and managing mental health within communities and can provide valuable insights for public health policies and interventions.

Given that anxiety and depression are mental health disorders and have a multidimensional nature, composite indicators are a natural methodological choice for assessing the propensity for anxiety and depression in children [15,37,38].

In particular, the use of composite indicators to assess the risk of depression and anxiety in children associated with exposure to adversity is an emerging area of research [19,39,40].

The regression and longitudinal analysis approach offers valuable information for understanding the multidimensionality of mental health over time [41–43]. However, it fails when the objective is to compare the mental health of populations from different regions, cities, and countries.

Another approach found in the literature for constructing composite mental health indicators is factor analysis [43–45]. This approach assigns higher weights to the sub-indicators most correlated to the phenomenon that isn't influenced by biases and errors of judgment. However, these weights may be incompatible with the theoretical framework.

## Childhood anxiety and depression propensity composite indicator

Whereas exposure to adverse childhood experiences elevates the risk of occurrence of anxiety and depression disorders [15,23] and several factors can generate adverse experiences in childhood [22], it is important to search for measures that demonstrate the influence of adverse childhood experiences on the propensity to anxiety and depression in children and adolescents.

Considering the context of adverse childhood experiences, this study adopts three dimensions for the construction of the composite indicator of propensity to anxiety and childhood depression. The proposed dimensions group and synthesize factors related to socioeconomic conditions, vulnerability, and risk/insecurity to represent exposure to adversity that affects children and adolescents.

The percentage of children and adolescents who have difficulty with anxiety and depression is adopted as the external variable. The external variable plays important roles in data analysis, such as moderating the relationships between variables, providing additional information for predictive models, and assisting in selecting variables [46].

## Adverse childhood experiences dimensions and factors

Going beyond the prevailing focus on child maltreatment, particularly sexual and physical abuse, in high-income countries [23], the adverse childhood experiences in this study address other factors that also increase the risk of mental disorders in children and adolescents in low and middle-income countries. Factors such as low socioeconomic status, discrimination, interparental conflict, parental mental and chronic illnesses, and exposure to violence or wars [24].

In this sense, the socioeconomic dimension incorporates factors related to the income level of the countries, the level of education of parents and children/adolescents, the possession of material goods, and health insurance. The vulnerability dimension aggregates factors related to the parent's perception of the evolution of living conditions and happiness and the presence of parents in the children's daily lives. The risk/insecurity dimension comprises factors related to domestic violence, living in violent places, performing domestic tasks or working in unfavorable conditions, and exposure to discrimination. The factors used in each dimension are presented in Table 1.

The factors included in each dimension were selected based on data availability and conceptual compatibility with the proposed dimensions defined based on the literature on adverse childhood experiences.

**Table 1. Analysis dimensions and their factors.**

| | |
|---|---|
| **Socioeconomic** | Average GDP per inhabitant; Average level of education of children/adolescents 0-No education, 1-Primary, 2-Secondary education (lower), 3-Secondary education (upper), 4-Technical education or higher; Mothers' average education level 0-No education, 1-Primary, 2-Secondary education (lower), 3-Secondary education (upper), 4-Technical education or higher; Average level of education of the fathers 0-No education, 1-Primary, 2-Secondary education (lower), 3-Secondary education (upper), 4-Technical education or higher; Mean age of the mothers interviewed in that country; Average age of the fathers interviewed from that country; Percentage of households interviewed that have some type of electricity; Percentage of households interviewed that have color or black and white TV; Percentage of households interviewed that have a refrigerator; Percentage of households interviewed that have a computer; Percentage of households interviewed that have a mobile phone; Percentage of households interviewed that have access to the internet; Percentage of children/adolescents in that country who have some type of health insurance; Percentage of mothers interviewed in that country who have some type of health insurance; Percentage of parents interviewed in that country who have some form of health insurance. |
| **Vulnerability** | Percentage of mothers in that country who believe that life has worsened compared to the previous year; Percentage of parents in that country who believe that life has worsened compared to the previous year; Percentage of mothers in that country who believe that life in the following year will be worse than the one they live in the current year; Percentage of fathers in that country who believe that life in the following year will be worse than the one they live in the current year; Average of the general unhappiness of mothers with life in that country 1-happy, 2-somehow happy, 3-neither happy, nor unhappy, 4-unhappy in some way, 5-very unhappy; Average of the fathers general unhappiness with life in that country 1-happy, 2-somehow happy, 3-neither happy nor unhappy, 4-unhappy in some way, 5-very unhappy; Percentage of biological fathers from that country who live with the child/adolescent in the same household; Percentage of biological mothers in that country who live with the child/adolescent in the same household; Percentage of biological fathers of the children/adolescents who are alive. |
| **Risk/insecurity** | Percentage of the population of children/adolescents who suffer aggressive or violent discipline by an adult family member; Average number of days the mothers consumed alcohol in the last month; Percentage of mothers interviewed who feel safe alone at home after dark; Percentage of parents surveyed who feel safe alone at home after dark; Percentage of mothers interviewed who feel safe walking the neighborhood streets alone after dark; Percentage of parents interviewed who feel safe walking the neighborhood streets alone after dark; Percentage of children/adolescents who needed to work in some paid service to help the family in the last week; Percentage of children/adolescents who performed some type of unhealthy, arduous or dangerous work in the last week; Percentage of children/adolescents who had to fetch water outside the home in the last week; Percentage of children/adolescents who needed to collect wood to use in the fire inside the house; Percentage of mothers who believe that children/adolescents need to be physically punished to be educated correctly; Percentage of mothers and/or fathers who were victims of some type of urban violence in the last 12 months; Percentage of mothers or fathers who felt personally discriminated against for some reason in the last year. |

In this study, the OWA operator was adopted to construct composite indicators, seeking to overcome the flaws present in other methods.

## OWA-based composite indicator

The OWA operator is a robust tool for constructing composite indicators [47]. It offers significant flexibility in adjusting the compensation levels between sub-indicators during the aggregation process [48].

The OWA operator was developed by [45] and introduced the possibility of biasing the composite indicator scores upwards or downwards. In other words, it allows the selection of the most significant sub-indicators for each decision unit's analysis. This enables decision-makers to emphasize the positive or negative aspects of the multidimensional phenomenon [49]. It should be noted that, in this case, the decision-maker is the one who builds the model.

The emphasis is also flexible regarding intensity, offering multiple solutions for representing the multidimensional phenomenon by adjusting the compensation level between sub-indicators [49,50].

Constructing a composite indicator using the ordered weighted average operator is quite simple and involves the following steps:

1. Normalizing the sub-indicators.

2. Transposing the matrix of normalized sub-indicators $\Omega_1, \Omega_2, \ldots, \Omega_\rho$.

3. Ordering the normalized sub-indicators in descending order.

4. Defining the weights $\beta_1, \beta_2, \ldots, \beta_\rho$ according to the ordered matrix.

5. Calculate the composite indicator using the weighted average.

The normalization of sub-indicators performed in the first step can be performed using the following expression:

$$\Omega_\lambda = \frac{\omega_\lambda - \min(\omega_\lambda)}{\max(\omega_\lambda) - \min(\omega)}$$

(1)

where $\omega_\lambda$ represents the value of the sub-indicator λ for the decision unit ω, and $\max(\omega_\lambda)$ and $\min(\omega_\lambda)$ are the maximum and minimum values of the sub-indicator λ across all decision units ψ.

After transposing the normalized sub-indicator matrix in the second step, the sub-indicators are ordered from highest to lowest in the third step. In the fourth step, the sub-indicators are weighted. Note that the weights $\beta$ depend on the intensity and direction of the emphasis. The choice of these parameters will define which lines of the matrix will be weighted with values different than zero.

For example, consider a multidimensional phenomenon with four sub-indicators. Then, assume that we want to emphasize the negative aspects of the multidimensional phenomenon. At this point, it is possible to establish a certain intensity of the negative emphasis. For example, it is possible to assign zero weight to the row of the matrix with the highest value and to emphasize the negative aspects slightly by assigning the weights $\left(\beta = \frac{1}{\rho}\right)$ to the remaining rows that satisfy the conditions $\beta_\rho \in [0, 1]$ and $\sum_{\lambda=1}^{\rho} \beta_\lambda = 1$.

Finally, the composite indicator is constructed in the fifth step using the following expression:

$$OWA(\Omega_1, \Omega_2, \ldots, \Omega_\rho) = \sum_{\lambda=1}^{\rho} \beta_\lambda \alpha_\lambda$$

(2)

where $\alpha_\lambda$ corresponds to the λ-th highest value among $\Omega_1, \Omega_2, \ldots, \Omega_\rho$, and the weights $\beta_\lambda$ satisfy the conditions $\beta_\lambda \in [0, 1]$ and $\sum_{\lambda=1}^{\rho} \beta_\lambda = 1$.

In short, after data normalization, the sub-indicators in descending order are weighted and then aggregated in a single score.

## Quality test

This section presents tests for explanatory power, discriminant power and proportion of outlier measures. These tests allow validating the quality and robustness of the results.

The explanatory power gauges the composite indicator's ability to encapsulate the essence of the multidimensional phenomenon. This is quantified using Spearman's correlation coefficient (ρ), Equation 3, which is robust for both normal and non-normal data distributions, even with outliers:

$$\rho = \frac{cov\left[K_{OWA_\omega}, K_{\epsilon_\omega}\right]}{\sigma_{k_{OWA_\omega}}\, \sigma_{K_{\epsilon_\omega}}}$$

(3)

where $\epsilon_\omega$ is the external variable, $cov\left[K_{OWA_\omega} K_{\epsilon_\omega}\right]$ is the covariance of the $OWA_\omega$ ranking and $OWA_\omega$ and the parameters $\sigma_{K_{OWA_\omega}}$ and $\sigma_{K_{\epsilon_\omega}}$ are the standard deviations of the $OWA_\omega$ and $\epsilon_\omega$ ranks, respectively.

Discriminant power measures the dispersion of the composite indicator's scores, Equation 4, providing a measure of the diversity of the information. It indicates the ease (or difficulty) for decision-makers to differentiate between decision units. Shannon's entropy index [51] is used to measure discriminant power:

$$H' = -\frac{1}{\ln \delta} \sum_{\omega=1}^{\delta} OWA_\omega \ln OWA_\omega$$

(4)

Where $H'$ represents Shannon's information diversity, and ln refers to the natural logarithm.

The proportion of outlier measurements assesses the compatibility of the composite indicator's scores with the external variable, Equation 5. It provides a measure of the quality of the decision units' measurements, allowing for the identification of outliers that should be analyzed individually. Mahalanobis distance [52] is used to measure the proportion of outlier measurements:

$$B = \frac{1}{\delta} \sum_{\omega=1}^{\delta} \begin{cases} 1 \ \textit{if } \psi^2 \le \eta_\omega^2 \\ 0 \ \textit{if } \psi^2 > \eta_\omega^2 \end{cases}$$

(5)

Where $\psi^2$ follows a chi-square distribution with degrees of freedom equal to the number of decision units, and $\eta^2$, the Mahalanobis distance, Equation 6, is obtained as follows:

$$\eta^2 = (OWA_\omega - \epsilon_\omega)^\nu \times \gamma^{-1} \times (OWA_\omega - \epsilon_\omega)$$

(6)

Where $\nu$ represents the matrix transpose, and $\gamma$ corresponds to the covariance matrix of $\epsilon_\omega$.

In short, the explanatory power measures the strength of the correlation between the composite indicator generated and an external variable that represents the phenomenon under study. The discriminant power represents the ease of distinguishing between the composite indicators generated for the decision units. Finally, the proportion of outlier measurements indicates that decision units presented results incompatible with the external variable, i.e., if a direct relationship is expected, the decision unit has a high composite indicated value and a low external variable value.

## Database

The databases used in this study, shown in Table 2, were obtained from UNICEF (https://mics.unicef.org/surveys) and are publicly available upon registration and access approval. These databases were extracted from a large global research project in low- and middle-income countries called MICS. In this study, we use data from the sixth round (MICS6),

**Table 2. Countries and data availability period.**

| Available data | Country |
|---|---|
| 2018–2019 | Guinea-Bissau, Algeria, Kiribati, Central Africa Republic, Republic of North Macedonia and Pakistan (Sindh) |
| 2019–2020 | Malawi, Palestine, Dominican Republic, Turks and Caicos Islands, Guyana and Argentina |
| 2022–2023 | Afghanistan and Yemen |
| 2020–2021 | Vietnam |
| 2021–2022 | Eswatini, Benin and Uzbekistan |
| 2017–2018 | Pakistan (Punjab), Ghana, and the Democratic Republic of Congo |
| 2019–2020 | Tuvalu, Kosovo, Pakistan (Balochistan), and Samoa |
| 2017 | Togo and Sierra Leone |
| 2018 | Gambia, Iraq, Suriname, Georgia, Lesotho, Mongolia, Costa Rica, Madagascar, and Montenegro |
| 2019 | Bangladesh, Chad, Cuba, Nepal, Pakistan (Khyber Pakhtunkhwa), Dominican Republic, Sao Tome and Principe, Belarus, Serbia, Honduras, Tonga, Turkmenistan and Zimbabwe |
| 2021 | Nigeria and Fiji |
| 2022 | Trinidad and Tobago, Comoros |
| 2023 | Kyrgyzstan and Tunisia |

conducted between 2018 and 2019 by the Ministry of Economy and Finance. MICS6 covers 54 countries (this data was downloaded in March 2024) and includes 72 questionnaires, 177 core indicators, and an average sample of 12,000 households, making it the largest edition.

Although its primary focus is women and children, several countries also administer the questionnaire to men. Trained fieldwork teams conduct face-to-face interviews with family members on various topics, primarily focusing on issues that directly affect the lives of children and women.

The Multiple Indicator Cluster Surveys survey is currently in its seventh edition (Round 7), which is yet to be completed in most countries. Therefore, this study utilized data from the sixth edition of the Multiple Indicator Cluster Surveys (Round 6), where data from 54 countries were available. As the surveys are conducted under the conditions of each country, the data provided refers to different periods collected between 2017 and 2023.

## Construction of composite indicators

The construction of propensity to childhood anxiety and depression composite indicator that seeks to reflect the importance of exposure to adverse experiences is conducted in two stages.

The first stage involves calculating a composite indicator for the three dimensions (socioeconomic, vulnerability, and risk/insecurity). These dimensions will subsequently be considered as sub-indicators of the composite indicator of propensity for anxiety and depression in childhood. The percentage of the population of children/adolescents who have difficulty in the field of anxiety and depression was adopted as an external variable.

The OWA was employed to calculate the sub-indicators representing these dimensions. The definition of the emphasis adopted will be supported by verifying the quality of the results of the composite indicators generated for each dimension. Tests are conducted to verify the explanatory power, discriminant power, and proportion of atypical measurements of the composite indicator to ensure its overall quality and robustness.

In the subsequent phase, a composite indicator is computed utilizing the Ordered Weighted Averaging (OWA) method, considering the sub-indicators derived for each dimension. Quality and robustness will also be verified through tests.

The methodological framework for constructing the composite indicator is represented in Fig 1.

In both proposed stages, the S-CI-OWA software is used to calculate the composite indicators through the OWA and to conduct quality tests on the results. The S-CI-OWA software is available at https://ci-owa.shinyapps.io/OWA-G1/.

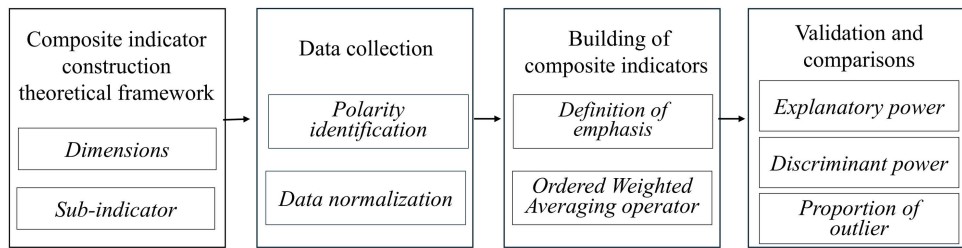

**Fig 1. Methodological framework.** The results provide a composite indicator encompassing the three dimensions in a single measure, indicating its validity as a representation of the multidimensional phenomenon under study.

### Propensity to childhood anxiety and depression in low-and middle-income countries

This section presents the construction process results for the composite indicator of propensity to childhood anxiety and depression in low- and middle-income countries.

Following data normalization, the emphasis was established. To this end, this study compared the results of quality tests applied to the composite indicators constructed to represent the dimensions using models without emphasis and with an emphasis of 25% on negative factors.

The quality test results for explanatory power, discriminating power, and the ratio of atypical measurements, as presented in Table 3, indicate no significant differences between the two models. The model without emphasis was adopted to preserve most information diversity.

To calculate the composite indicator of propensity to anxiety and depression in children, the results generated for the composite indicators of the dimensions become the sub-indicators. Table 4 shows the values.

The average value of the composite indicator obtained was 0.52. The country with the highest value for the composite indicator was Central Africa Republic (0.77), and the country with the lowest value was Georgia (0.24). See the value for all countries in data S1 Table. This result indicates that among the countries analyzed, the Central Africa Republic has the highest propensity for childhood anxiety and depression and Gerorgia the lowest.

Looking at the values obtained for the dimensions, considering all countries in the analysis, the sub-indicators obtained average values of 0.57 for socioeconomics, 0.61 for vulnerability, and 0.50 for risk/insecurity.

The vulnerability dimension generated the composite indicator with the highest average value. This means that when we analyze the dimensions separately concerning the percentage of the population of children and adolescents who have difficulty in the field of anxiety and depression, the vulnerability factors generate, on average, the worst result. Considering

**Table 3. Quality tests of the dimension indicators.**

| Dimension | Test | No emphasis | 25% negative emphasis |
|---|---|---|---|
| **Risk/ Insecurity** | Explanatory Power: | 0.399 | 0.379 |
| | Ratio of atypical measurements: | 0.037 | 0.056 |
| | Discriminating power: | 5.755 | 5.755 |
| **Vulnerability** | Explanatory Power: | 0.218 | 0.224 |
| | Ratio of atypical measurements: | 0.019 | 0.037 |
| | Discriminating power: | 5.755 | 5.755 |
| **Socioeconomic** | Explanatory Power: | 0.303 | 0.302 |
| | Ratio of atypical measurements: | 0.056 | 0.056 |
| | Discriminating power: | 5.755 | 5.755 |

**Table 4. Descriptive statistics of the sub-indicators obtained for the dimensions and the composite indicator of propensity to childhood anxiety and depression.**

|         | Socioeconomic | Vulnerability | Risk/Insecurity | Composite Indicator |
|---------|---------------|---------------|-----------------|---------------------|
| Min.    | 0.22          | 0.33          | 0.24            | 0.24                |
| Max.    | 0.87          | 0.93          | 0.74            | 0.77                |
| Mean    | 0.57          | 0.61          | 0.50            | 0.52                |
| Median  | 0.57          | 0.61          | 0.49            | 0.53                |
| Std Dev | 0.17          | 0.13          | 0.10            | 0.13                |

the total group of countries under analysis, this result indicates that the vulnerability dimension requires the most attention. In this sense, public policies for social protection, investment in infrastructure, and public security should be given priority.

The standard deviation and the maximum and minimum values indicate a large amplitude and dispersion of the values obtained for the dimensions and composite indicator country.

Considering the wide range of results, the analysis considering the countries by continent can make it possible to see if there are differences between the results by geographic location and which sub-indicators have the greatest influence on each continent. This can help the formulation of more appropriate public policies for each region.

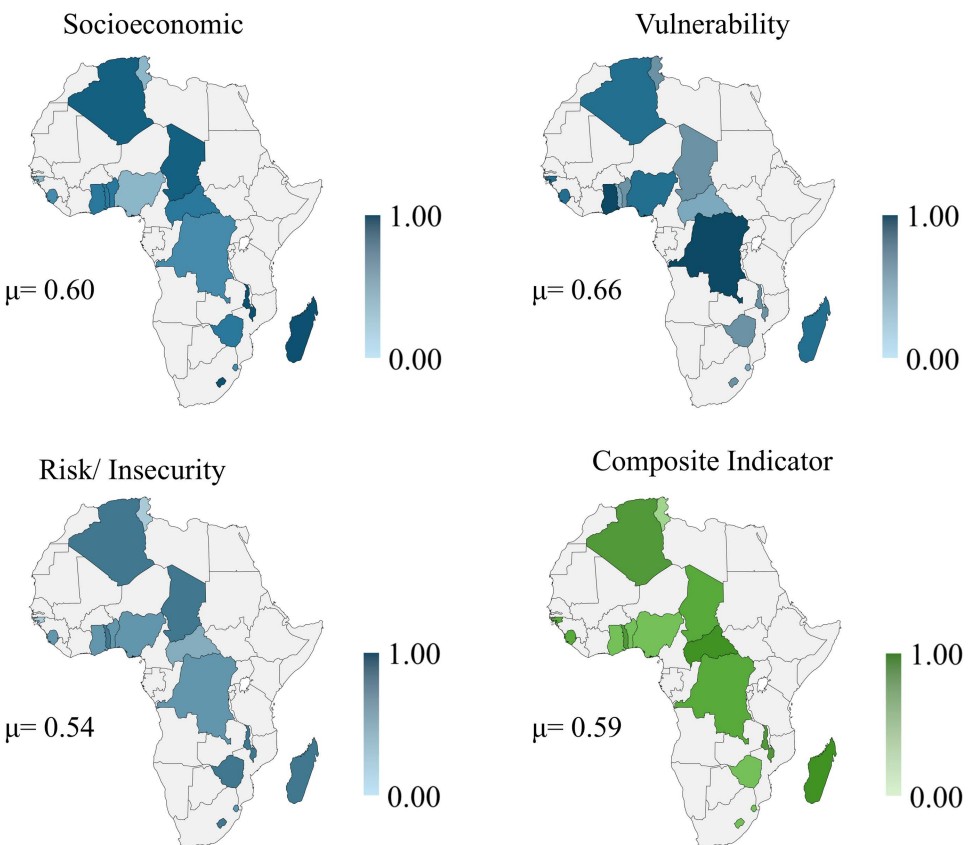

**Fig 2. Results map of countries in Africa.** Republished from [53] under a CC BY license, with permission from Minas Giannekas, original copyright Copyright © 2025 MapChart.

Fig 2 shows the findings about Africa. The composite indicator has an average value of 0.59. The Central African Republic had the highest value for the composite indicator (0.77), and Tunisia had the lowest value (0.37). The highest average value of the sub-indicators is in the vulnerability dimension (0.64), followed by socioeconomic (0.58) and risk/insecurity (0.51) with the lowest value. See the value for African countries in the S3 Table of data.

These results indicate that children and adolescents from countries on the African continent have a moderate to high propensity to childhood anxiety and depression. Public policies that promote the fight against poverty, social inclusion, and investments in infrastructure can help reduce childhood anxiety and depression in these countries.

Latin America and the Caribbean countries obtained an average value for the composite indicator of 0.40. Costa Rica had the highest value for the composite indicator (0.49), and Turks and Caicos Islands had the lowest value (0.26). The results are presented in Fig 3 and indicate that the socioeconomic sub-indicator (0.63) has a higher average value, followed by vulnerability (0.57) and risk/insecurity (0.52). See the value for American countries in the S4 Table.

In these countries, the propensity to childhood anxiety and depression tends to be moderate to low. Actions related to socioeconomic factors will have greater power in combating anxiety and childhood depression related to exposure to adversity. Public policies for access to education, investment in health, and qualification of workers are examples of actions that can generate superior results in combating childhood anxiety and depression in the American countries analyzed.

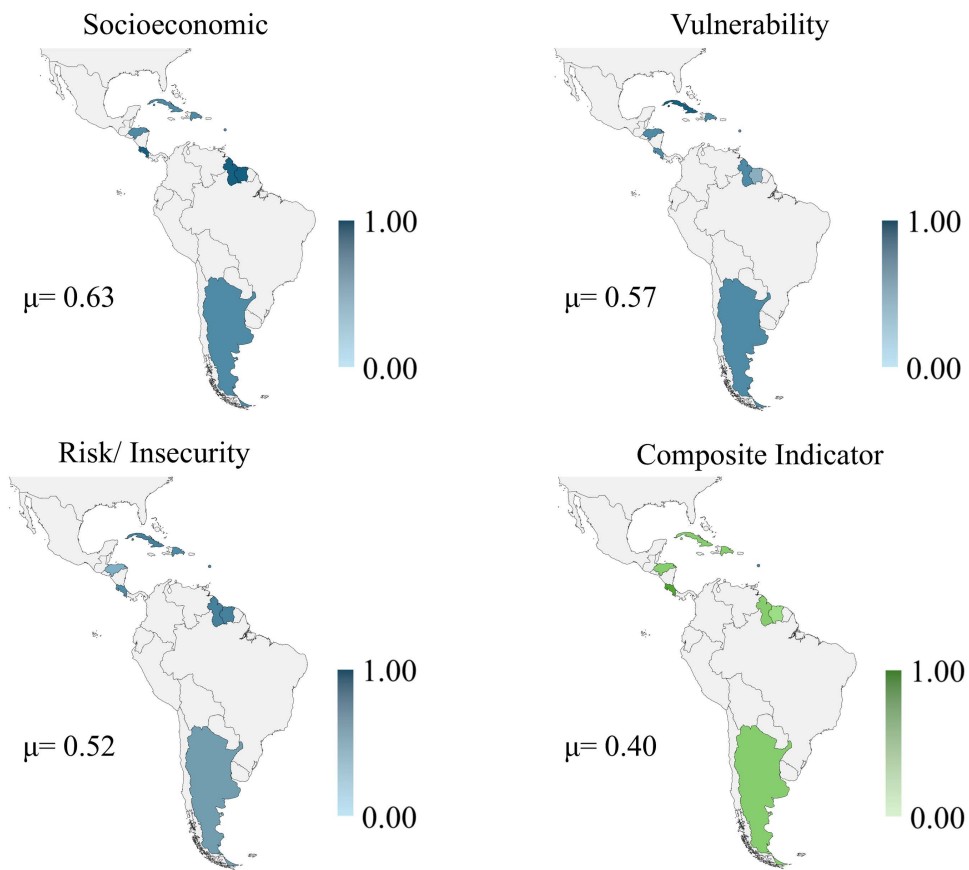

**Fig 3. Results maps for Latin American and Caribbean countries.** Republished from [53] under a CC BY license, with permission from Minas Giannekas, original copyright Copyright © 2025 MapChart.

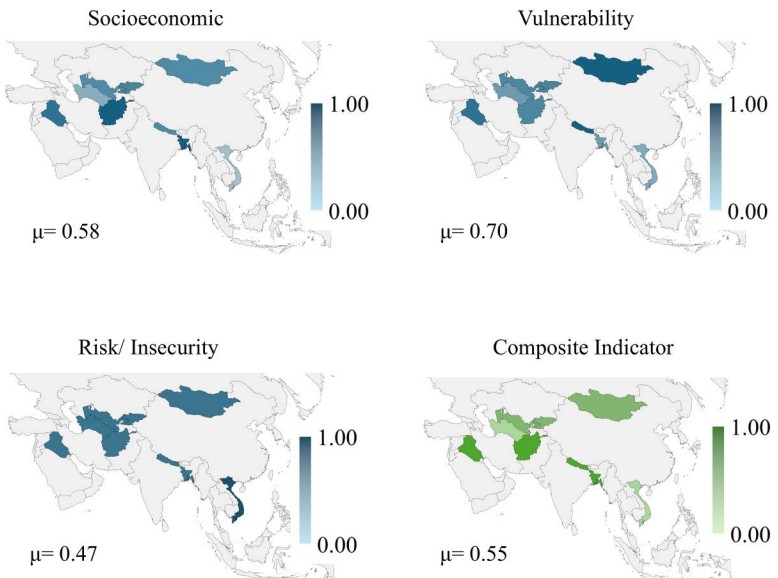

**Fig 4. Results map for countries in Asia.** Republished from [53] under a CC BY license, with permission from Minas Giannekas, original copyright Copyright © 2025 MapChart.

Asian countries obtained an average value of the composite indicator of 0.55. Afghanistan had the highest value for the composite indicator (0.75), and Vietnam had the lowest value (0.30). The results shown in Fig 4 indicate that the sub-indicators of vulnerability have a higher relative average weight at 0.70. The sub-indicators of socioeconomics (0.58) and risk/ insecurity (0.47) had lower weights. See the value for Asia countries in data S5 Table.

In the Asian countries under analysis, the propensity to childhood anxiety and depression tends to be moderate to high. Improvements in vulnerability risk and insecurity factors can contribute more to preventing these difficulties. Children from the Asian countries under analysis have a moderate to low propensity to childhood anxiety and depression. The fight against childhood anxiety and depression can be more effective if it prioritizes public policies on actions against poverty, social inclusion, and investments in infrastructure.

The countries of Europe obtained an average value for the composite indicator of 0.43. The Republic of North Macedonia obtained the highest value for the composite indicator (0.66), and Georgia obtained the lowest value (0.24). Fig 5 shows that the vulnerability sub-indicator (0.58) has the highest average weight compared to socioeconomic (0.47) and risk/insecurity (0.50) sub-indicators. See the value for European countries in data S6 Table.

Public policy actions focused on factors related to vulnerability can contribute more effectively to combating the propensity to childhood anxiety and depression in the European countries analyzed.

The countries of Oceania, as shown in Fig 6, obtained an average value for the composite indicator of 0.40. Kiribati was the country with the highest value for the composite indicator (0.70), and Tonga had the lowest value (0.27). The dimension related to risk/insecurity factors obtained the highest average value (0.48), followed by socioeconomic dimension (0.42) and vulnerability (0.42), with the lowest value. See the value for Oceania countries in data S6 Table.

Combating propensity to childhood anxiety and depression in Oceania countries must encompass all dimensions, with emphasis on risk and insecurity. Considering this result, actions to increase public safety and policies to combat domestic violence, child labor, and discrimination can generate a greater reduction in the indicator of propensity to anxiety and depression in children and adolescents.

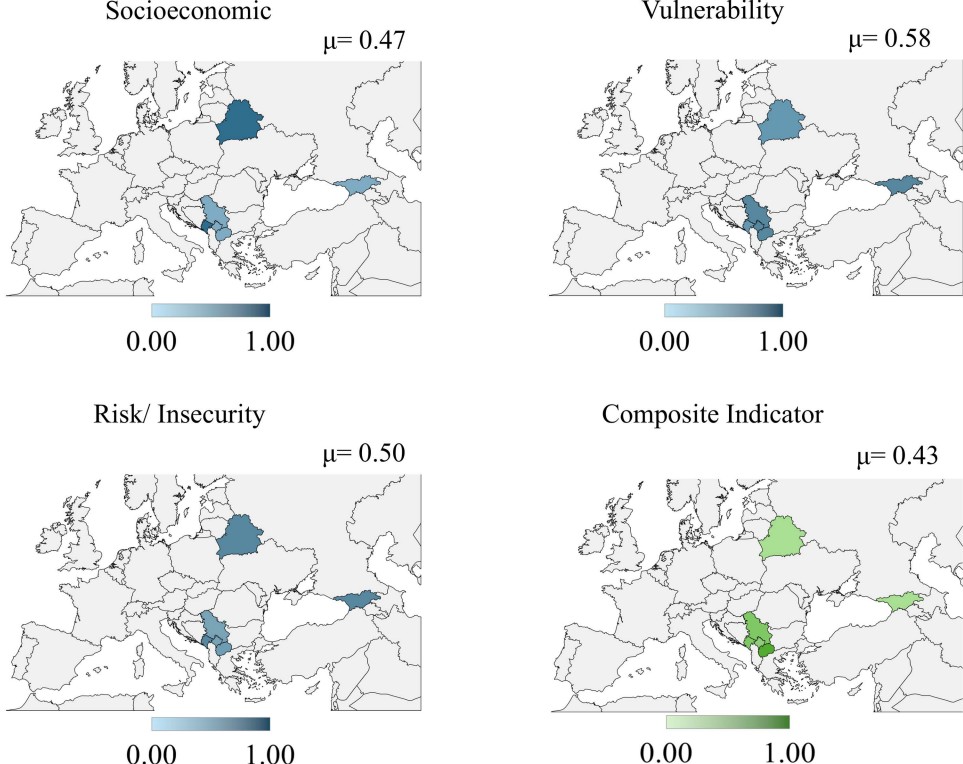

**Fig 5. Results map European countries.** Republished from [53] under a CC BY license, with permission from Minas Giannekas, original copyright Copyright © 2025 MapChart.

The distribution of the importance of each dimension in the composite indicator of propensity to anxiety and childhood depression is shown in Fig 7. It is possible to observe that the influence of each dimension changes between continents. These findings align with the adversity factors children and adolescents face on each continent. See the value for all countries in data S8 Table.

Considering the composite indicator generated by the three dimensions, the socioeconomic dimension has a higher proportion in relation to the other dimensions in African countries. The vulnerability dimension represented the highest proportion in American, Asian, and, especially, European countries. For the countries belonging to Oceania, the risk/insecurity dimension had a higher proportion.

The differences found in the proportions of the dimensions between continents reinforce the difference in the context existing in each continent and the need for specific public policies. Using Europe as an example, where more than 50% of the value obtained for the composite indicator of propensity to anxiety and depression is attributed to the vulnerability dimension, it is expected that actions aimed at the social well-being of families will have a greater effect on reducing propensity to childhood anxiety and depression than actions to combat poverty and public security.

Below are the results of the composite indicator quality test. The results in Fig 8 demonstrate that the composite indicator has a satisfactory explanatory power. The external variable has a positive correlation with the composite indicator (rho = 0.53), socioeconomic dimension (rho = 0.30), vulnerability dimension (rho = 0.22), and risk/ insecurity dimension (rho = 0.40). This correlation signals that the composite indicator captures the difficulties encountered by children and adolescents in the field of anxiety and depression.

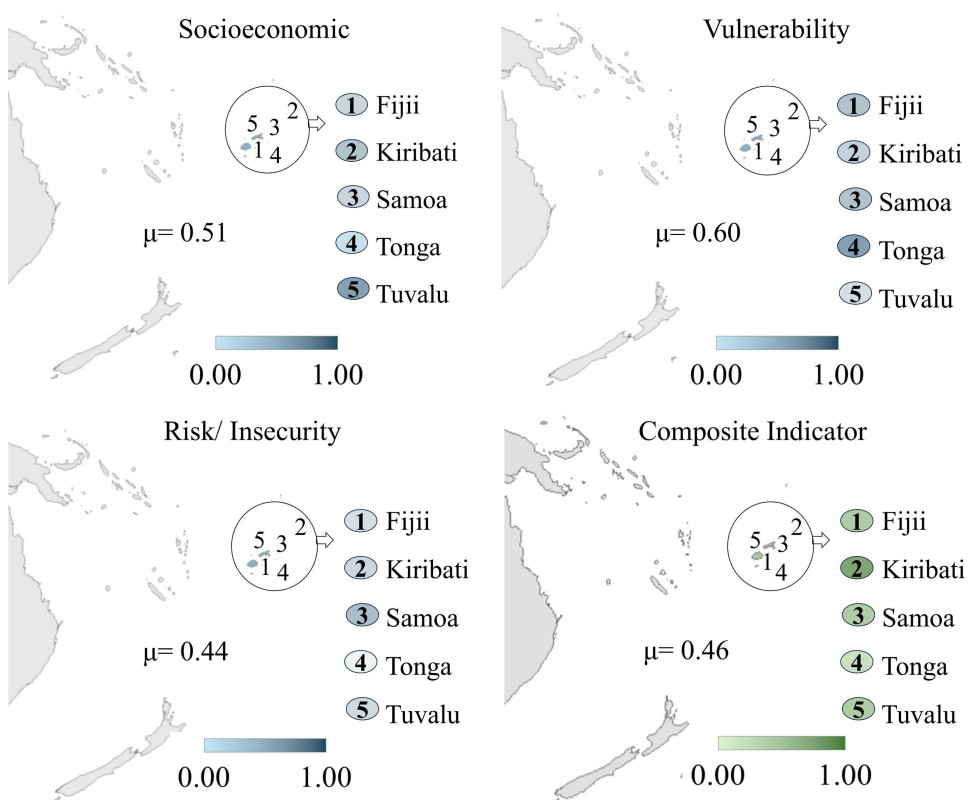

**Fig 6. Results maps for Oceania countries.** Republished from [53] under a CC BY license, with permission from Minas Giannekas, original copyright Copyright © 2025 MapChart.

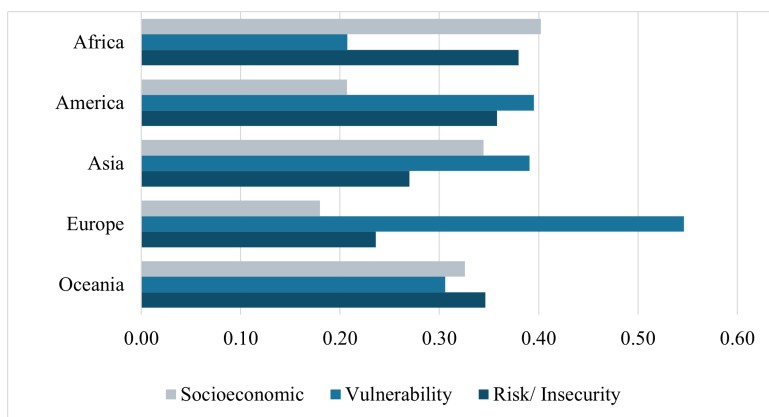

**Fig 7. Proportion of dimensions in the composite indicator by continent.**

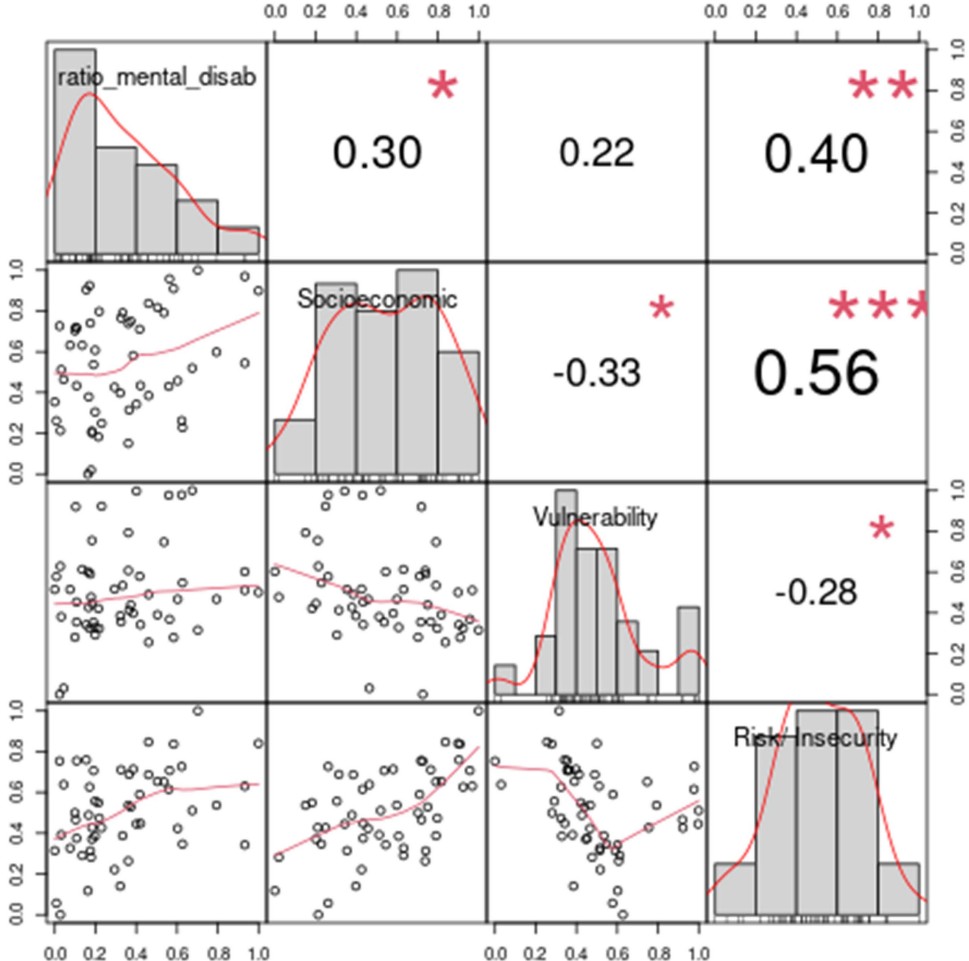

**Fig 8. Correlation results of the dimensions and the OWA CI with the external variable.** The results indicate that the higher indicator, the greater the difficulty faced by children and adolescents in the field of anxiety and depression. This result corroborates the results found in the literature that relate adverse childhood experiences with anxiety and depression [13,43,54].

Discriminating power indicates the ease of distinguishing the scores of the composite indicator, being evidence of quality and reliability. The histogram in Fig 9 suggests that the country's scores are reasonably distributed among the score classes, making it easier for the decision-maker to differentiate the countries. This ease of differentiation between the results obtained by the countries can be seen in the maps presented.

The proportion of atypical measurements is the latest evidence of the quality and reliability of the composite indicator adopted in this study. The graph of bivariate outliers in Fig 10 shows that only two countries are outside the orange ellipse. This indicates the low existence of atypical measurements, a proportion of 0.037. This means that two countries (Afghanistan and Montenegro) had scores of the composite indicator of propensity to anxiety and childhood depression that are not consistent with incompatible with the percentage of the population of children and adolescents who have difficulties in the field of anxiety and depression. In other words, given the value obtained for the composite indicator of childhood anxiety and depression, Afghanistan and Montenegro should have smaller proportions of children and adolescents with difficulties in the field of anxiety and depression. The presence of armed conflict and other factors not considered may cause these countries to present atypical results.

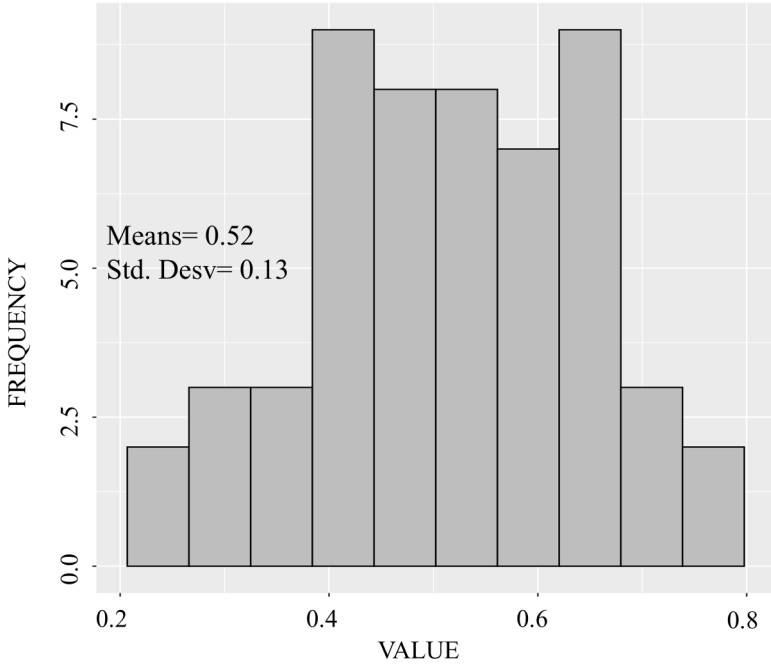

**Fig 9. Discriminating power.**

The lower the proportion of atypical measurements, the better the composite indicator represents the phenomenon under analysis.

The findings provide an initial profile of the propensity to childhood anxiety and depression related to exposure to childhood adversity in low- and middle-income countries. Appropriately utilizing these findings in formulating public policies necessitates a comprehensive examination of the unique circumstances pertinent to each nation.

## Conclusions

The World Health Organization reports that mental health conditions, such as depression, anxiety, and other emotional disorders, account for 16% of the global disease and disability burden among adolescents aged 10–19. Numerous studies have concentrated on developing tools and scales to assess mental health conditions. In particular, the use of composite indicators to evaluate the risk of anxiety and depression in children and adolescents, considering multidimensional factors such as exposure to adversity, is an expanding area of research.

In this context, the study proposed the development of a composite indicator to assess the propensity for childhood anxiety and depression based on dimensions that capture childhood and adolescent exposure to adversity. The OWA was employed to construct the composite indicator, and several quality tests were conducted to validate its results.

The generated composite indicator reveals that, on average, countries in Africa and Asia show a higher propensity for childhood anxiety and depression. Conversely, the Americas have the lowest average propensity for these mental health conditions.

The breakdown of the composite indicator results by dimensions shows that each dimension has varying degrees of influence in each continent. In Africa, the socioeconomic dimension has the greatest impact, while vulnerability is the most significant factor in the Americas, Asia, and Europe. These findings provide a valuable tool for guiding the formulation of

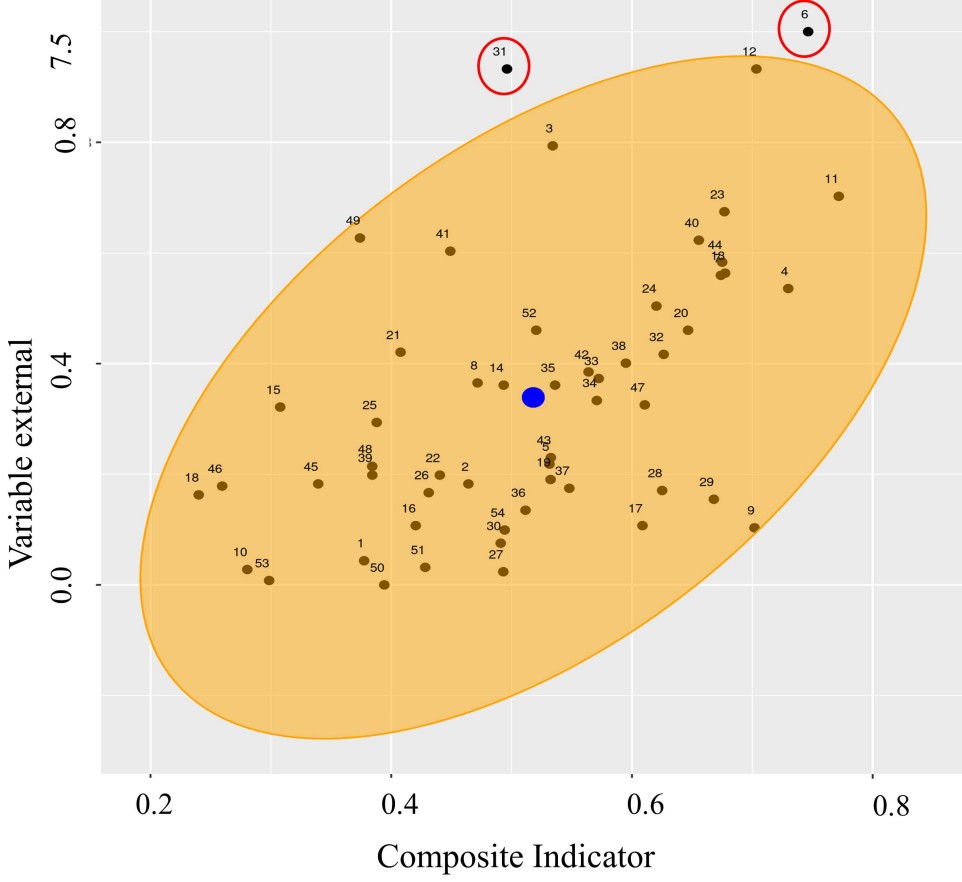

**Fig 10. Atypical measurements.**

effective public policies aimed at mitigating factors that contribute to anxiety and depression in children and adolescents, potentially leading to significant social and economic benefits.

The validation of the results through quality tests reinforces confidence in the direction indicated by the findings, enhancing the decision-making process when dealing with multidimensional phenomena.

While the quality tests indicate that the composite indicator is reliable, future research concerning adding other tests could improve its reliability. This study serves as a foundation for future research to refine the sub-indicators and expand the scope of countries under analysis.

The lack of data availability is one of the limitations of this study. The use of existing research data, such as MICS, may be affected by sampling or response biases, and data for many countries are lacking.

Another point of attention is that correlation analyses do not imply causality, and other factors not considered in the model can influence the results.

Future studies should seek to expand and improve the models and techniques for analyzing and validating results, as well as improvements in the data used and deepening the application of results in elaborating public policies. Another point of interest for future research is to explore the composite indicator's validity in comparing countries considered developed with underdeveloped countries.

## Supporting information

**S1 Table. database.**
(XLSX)

**S2 Table. Descriptive statistics.**
(XLSX)

**S3 Table. Date America countries.**
(XLSX)

**S4 Table. Date Africa countries.**
(XLSX)

**S5 Table. Date Asian countries.**
(XLSX)

**S6 Table. Date Europe countries.**
(XLSX)

**S7 Table. Date Oceania countries.**
(XLSX)

**S8 Table. Sub-indicators proportions.**
(XLSX)

## Author contributions

**Data curation:** Ariane Silva.

**Formal analysis:** Angélica C. G. Santos.

**Methodology:** Matheus Libório.

**Supervision:** Marcos Flávio S. V. D'Angelo.

**Validation:** Hasheem Mannan.

**Writing – original draft:** Angélica C. G. Santos.

**Writing – review & editing:** Cristiane Neri Nobre.

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
