## [Decision Letter · Decision Letter 0]

25 Feb 2025

PONE-D-24-60103Propensity to Childhood Anxiety and Depression Due to Exposure to Adversity: A Multidimensional ConstructPLOS ONE

Dear Dr. Mannan,

Thank you for submitting your manuscript to PLOS ONE. After careful consideration, we feel that it has merit but does not fully meet PLOS ONE’s publication criteria as it currently stands. Therefore, we invite you to submit a revised version of the manuscript that addresses the points raised during the review process.

**ACADEMIC EDITOR: ** Dear Authors,

Thank you for submitting your manuscript to Plos One.

Three Reviewers and the Academic Editor have evaluated your investigation.

Please, respond to all the comments below and make the necessary modifications into your manuscript.Thank you again for your patience and interest in Plos One.

We look forward to receiving your revised manuscript.

Kind regards,

Javier Fagundo-Rivera, PhD

Academic Editor

PLOS ONE

Journal Requirements:

2. We note that Figures 2-5 in your submission contain map/satellite images which may be copyrighted. All PLOS content is published under the Creative Commons Attribution License (CC BY 4.0), which means that the manuscript, images, and Supporting Information files will be freely available online, and any third party is permitted to access, download, copy, distribute, and use these materials in any way, even commercially, with proper attribution. For these reasons, we cannot publish previously copyrighted maps or satellite images created using proprietary data, such as Google software (Google Maps, Street View, and Earth). For more information, see our copyright guidelines: http://journals.plos.org/plosone/s/licenses-and-copyright.

 a. You may seek permission from the original copyright holder of Figure(s) [#] to publish the content specifically under the CC BY 4.0 license. 

3. We note you have included a table to which you do not refer in the text of your manuscript. Please ensure that you refer to Table 2 in your text; if accepted, production will need this reference to link the reader to the Table.

**Additional Editor Comments:**

Dear Authors,

Thank you for submitting your manuscript to Plos One.

Three Reviewers and the Academic Editor have evaluated your investigation.

Please, respond to all the comments and make the necessary modifications into your manuscript.

1)While the methodology is robust, the description of the OWA methods and quality tests could be simplified to enhance accessibility and clarity in the text for a broader audience.

1.1) It is crucial to provide a more detailed justification for the selection of the factors included in each dimension (socioeconomic, vulnerability, and risk/insecurity). The rationale for why these factors are the most relevant should be explained, along with how they relate to childhood anxiety and depression based on the existing literature.

2)While the manuscript presents descriptive results by continent, it identifies countries with atypical measurements (outliers). The analysis could be enriched by discussing the possible reasons for these anomalies.

2.1) A more in-depth analysis of the implications for public policy of these findings is necessary, and its implications in those specific contexts. What specific public policies could be implemented in each region to address the identified risk factors? How do these results compare to other studies in the literature?

3)It is important to explicitly state the study’s limitations, such as its reliance on existing survey data (MICS), which may be subject to sampling or response biases.

3.1) Additionally, it should be acknowledged that correlation does not imply causation and that other factors not considered in the model may be influencing the results.

4)Finally, it would be beneficial to mention the need for future research to explore the validity of the composite indicator across different cultural and socioeconomic contexts.

Reviewers' comments:

Reviewer's Responses to Questions

**Comments to the Author**

1. Is the manuscript technically sound, and do the data support the conclusions?

Reviewer #1: Yes

Reviewer #2: Partly

Reviewer #3: Yes

2. Has the statistical analysis been performed appropriately and rigorously? 

Reviewer #1: Yes

Reviewer #2: I Don't Know

Reviewer #3: Yes

3. Have the authors made all data underlying the findings in their manuscript fully available?

Reviewer #1: Yes

Reviewer #2: No

Reviewer #3: No

4. Is the manuscript presented in an intelligible fashion and written in standard English?

Reviewer #1: Yes

Reviewer #2: No

Reviewer #3: Yes

5. Review Comments to the Author

**Reviewer #1: **

• Abstract: "The literature does not present a consensus about the composition of indicators...” should be corrected to “The literature does not present a consensus on the composition of indicators.”

• Introduction: "Researchers estimate that approximately 6.5% of children and adolescents suffer from anxiety, while 2.6% are affected by depression” could be rephrased as “Approximately 6.5% of children and adolescents are estimated to suffer from anxiety, while 2.6% are affected by depression (Polanczyk et al., 2015).”

• Redundancy: The phrase “Studies primarily focus on the consequences observed in adulthood” is repeated multiple times throughout the manuscript. Consider consolidating these mentions for conciseness.

• Methodology Section: Some sentences explaining the OWA operator and validation tests are lengthy and complex. Simplifying these sentences would enhance reader comprehension.

• Terminology Consistency: Maintain consistent terminology throughout the manuscript. For instance, use “mental health conditions” or “anxiety and depression” uniformly unless contextually necessary to switch.

**Reviewer #2: **

Thank you for the privilege of reading your article.

This article details the synthesis and validation of a composite indicator to assess and predict the risk of anxiety and depression in children and adolescents in different countries based on the findings of the multiple indicator cluster surveys collected by UNICEF.

The opening of section 2 feels somewhat repetitive on reading and it is not entirely clear why it exists as a separate section from the introduction.

In section 3, line 269 the sentence is incomplete - Research on adverse childhood experiences has predominantly focused … (on what?). The figure 7 needs to be labelled clearly.

In section 2.2 it is not clear who the decision makers are, and what values of β were used for the construction of the composite indicator in this particular study. In section 4 it is offered that the sub indicators have similar relative weights. However it is not clear in this case what weights have been assigned to the individual factors in the different dimensions detailed in table 1 by the decision makers - or are the sub indicators unweighted averages of the the factors constituting them? This is revisited again in Figure 6 and it is not clear whether the perception of the relative importance of each dimension is based on the decisions of expert human decision makers or derived by deep learning algorithms in correlation with the external variables? Please clarify how this composite indicator can be used by other researchers and health care systems without this knowledge.

The article is thorough in defining the scope of the study. The introduction is quite long and circles back to the same point several times. The use of a OWA-CI is explained elaborately except for defining the decision maker. The study is useful to assess the need for intervention to protect the mental health of children and adolescents in areas where there is poor recognition of mental health issues. I wish the authors the very best with their future endeavours.

**Reviewer #3: **

Line 343: Please add a detailed explanation about the external variable because it is included up to the correlation analysisFor Figures 2,3,4, and 5, mention about the μ (average value of CI) to be self-explanatoryFor countries of Oceania, there is no figure and it should be added to be consistent with other regions.Line 295 and 399: the spelling of “Viet Nan” should be revised to “Vietnam”Line 427: what does “a little more” mean for socioeconomic factors, and risk and insecurity? This word is not precise and not measurable. So, it is suggested to change.Line 433: In African region, the most relevant sub-indicator is shown as the socioeconomic dimension. Although it is acceptable in comparison with other regions, risk/insecurity is a dimension with the highest proportion in Africa according to lines 376 and 377 (risk/insecurity occupied 41% followed by socioeconomic dimension (37%)), and Figure 6. Therefore, for line 433, not only the socioeconomic dimension but also risk/insecurity is needed to show a comparison with other regions.

6. PLOS authors have the option to publish the peer review history of their article (what does this mean? ). If published, this will include your full peer review and any attached files.

**Do you want your identity to be public for this peer review?** For information about this choice, including consent withdrawal, please see our Privacy Policy .

Reviewer #1: No

Reviewer #2: No

Reviewer #3: **Yes: ** May Soe Aung

---

## [Author Response · Author response to Decision Letter 1]

19 May 2025

PONE-D-24-60103

Propensity to Childhood Anxiety and Depression Due to Exposure to Adversity: A Multidimensional Construct

Javier Fagundo-Rivera, PhD

Dear Editor,

Thank you for your attention and time in dealing with our manuscript.

Fulfilling the comments sent, all issues raised have been attended to and properly addressed. The issues appointed to the manuscript have been highlighted and reproduced in the letter below.

Kind regards, 

The authors

##################Journal Requirements:##################

All PLOS ONE’s style requirements were attended.

We note that Figures 2-5 in your submission contain map/satellite images which may be copyrighted.

The maps in figures were powered by Microsoft Bing in excel program.

We note you have included a table to which you do not refer in the text of your manuscript. Please ensure that you refer to Table 2 in your text; if accepted, production will need this reference to link the reader to the Table.

The indication of table 2 has been added to the text.

Additional Editor Comments:

1) While the methodology is robust, the description of the OWA methods and quality tests could be simplified to enhance accessibility and clarity in the text for a broader audience.

Thank you for the suggestion. In this regard, we've expanded the OWA description, making it clearer and more accessible to a wider audience.:

[…] Constructing a composite indicator using the ordered weighted average operator is quite simple and involves the following steps:

Normalizing the sub-indicators.

Transposing the matrix of normalized sub-indicators \mathrm{\Omega}_1,\mathrm{\Omega}_2,\ldots,\mathrm{\Omega}_\rho.

Ordering the normalized sub-indicators in descending order.

Ordering the normalized sub-indicators in descending order.

Defining the weights \beta_1,\beta_2,\ldots,\beta_\rho according to the order matrix.

Calculate the composite indicator using the weighted average.

The normalization of sub-indicators performed in the first step can be performed using the following expression:

\mathrm{\Omega}_\lambda=\frac{\omega_\lambda-min\left(\omega_\lambda\right)}{max\left(\omega_\lambda\right)-min\left(\omega\right)} (1)

where \omega_\lambdarepresents the value of the sub-indicator λ for the decision unit ω, and max\left(\omega_\lambda\right) and min\left(\omega_\lambda\right) are the maximum and minimum values of the sub-indicator λ across all decision units ψ.

After transposing the normalized sub-indicator matrix in the second step, the sub-indicators are ordered from highest to lowest in the third step. In the fourth step, the sub-indicators are weighted. Note that the weights \beta depend on the intensity and direction of the emphasis. The choice of these parameters will define which lines of the matrix will be weighted with values different than zero.

For example, consider a multidimensional phenomenon with four sub-indicators. Then, assume that we want to emphasize the negative aspects of the multidimensional phenomenon. At this point, it is possible to establish a certain intensity of the negative emphasis. For example, it is possible to assign zero weight to the row of the matrix with the highest value and to emphasize the negative aspects slightly by assigning the weights \left(\beta=\frac{1}{\rho}\right) to the remaining rows that satisfy the conditions \beta_\rho\in\left[0,1\right] and \sum_{\lambda=1}^{\rho}\beta_\lambda=1.

Finally, the composite indicator is constructed in the fifth step using the following expression:

OWA\left(\mathrm{\Omega}_1,\mathrm{\Omega}_2,\ldots,\mathrm{\Omega}_\rho\right)=\sum_{\lambda=1}^{\rho}\beta_\lambda\alpha_\lambda (2)

where \alpha_\lambda corresponds to the λ-th highest value among \mathrm{\Omega}_1,\mathrm{\Omega}_2,\ldots,\mathrm{\Omega}_\rho, and the weights \beta_\lambda satisfy the conditions \beta_\lambda\in\left[0,1\right] and \sum_{\lambda=1}^{\rho}\beta_\lambda=1.

In short, after data normalization, the sub-indicators in descending order are weighted and then aggregated in a single score.

[…]

1.1) It is crucial to provide a more detailed justification for selecting the factors included in each dimension (socioeconomic, vulnerability, and risk/insecurity). The rationale for why these factors are the most relevant should be explained, along with how they relate to childhood anxiety and depression based on the existing literature.

Thank you for the suggestion. The position of the text sections has been modified, and some adjustments have been made to the text to clarify the relationship between the proposed dimensions and the propensity for childhood anxiety and depression. See the adjustments made to lines 127-231.

2) While the manuscript presents descriptive results by continent, it identifies countries with atypical measurements (outliers). The analysis could be enriched by discussing the possible reasons for these anomalies.

Thank you for the suggestion. The following explanations have been included to consider this suggestion:

“[…] the proportion of outlier measurements indicates that decision units presented results incompatible with the external variable, i.e., if a direct relationship is expected, the decision unit has a high composite indicated value and a low external variable value”. (lines 313-315)

“[…] In other words, given the value obtained for the composite indicator of childhood anxiety and depression, Afghanistan and Montenegro should have smaller proportions of children and adolescents with difficulties in the field of anxiety and depression. The presence of armed conflict and other factors not considered may cause these countries to present atypical results”. (line 543-553)

2.1) A more in-depth analysis of the implications for public policy of these findings is necessary, and its implications in those specific contexts. What specific public policies could be implemented in each region to address the identified risk factors? How do these results compare to other studies in the literature?

Thank you for the suggestion. Some indications have been added of what type of public policy would be more effective in each contingent.

“These results indicate that children and adolescents from countries on the African continent have a moderate to high propensity to childhood anxiety and depression. Public policies that promote the fight against poverty, social inclusion, and investments in infrastructure can help reduce childhood anxiety and depression in these countries.” (lines 412-415)

” In these countries, the propensity to childhood anxiety and depression tends to be moderate to low. Actions related to socioeconomic factors will have greater power in combating anxiety and childhood depression related to exposure to adversity. Public policies for access to education, investment in health, and qualification of workers are examples of actions that can generate superior results in combating childhood anxiety and depression in the American countries analyzed.” (438-443)

“In the Asian countries under analysis, the propensity to childhood anxiety and depression tends to be moderate to high. Improvements in vulnerability risk and insecurity factors can contribute more to preventing these difficulties. Children from the Asian countries under analysis have a moderate to low propensity to childhood anxiety and depression. The fight against childhood anxiety and depression can be more effective if it prioritizes public policies on actions against poverty, social inclusion, and investments in infrastructure.” (lines 452-457)

“Public policy actions focused on factors related to vulnerability can contribute more effectively to combating propensity to childhood anxiety and depression in the European countries analyzed.” (lines 468-470)

“Combating propensity to childhood anxiety and depression in Oceania countries must encompass all dimensions, with emphasis on risk and insecurity. Considering this result, actions to increase public safety and policies to combat domestic violence, child labor, and discrimination can generate a greater reduction in the indicator of propensity to anxiety and depression in children and adolescents.” (lines 480-484)

3) It is important to explicitly state the study’s limitations, such as its reliance on existing survey data (MICS), which may be subject to sampling or response biases.

Thank you for the comment. The limitations of the study were included as follows:

” The lack of data availability is one of the limitations of this study. The use of existing research data, such as MICS, may be affected by sampling or response biases, and data for many countries are lacking.” (lines 591-593)

3.1) Additionally, it should be acknowledged that correlation does not imply causation and that other factors not considered in the model may be influencing the results.

Thank you for the comment. This was included to address this suggestion:

“Another point of attention is the fact that correlation analyses do not imply causality and other factors not considered in the model can influence the results.” (lines 594-595)

4) Finally, it would be beneficial to mention the need for future research to explore the validity of the composite indicator across different cultural and socioeconomic contexts.

Thank you for the suggestion. Please, see the text included below:

“Future studies should seek to expand and improve the models and techniques for analyzing and validating results, as well as improvements in the data used and deepening the application of results in the elaboration of public policies. Another point of interest for future research is to explore the validity of the composite indicator in comparing countries considered developed with countries considered underdeveloped.” (lines 596-601)

##################Reviewer #1: ##################

Abstract: "The literature does not present a consensus about the composition of indicators...” should be corrected to “The literature does not present a consensus on the composition of indicators.”

Thank you for the suggestion. The suggested correction has been made (lines 27-28)

“The literature does not present a consensus on the composition of indicators.”

Introduction: "Researchers estimate that approximately 6.5% of children and adolescents suffer from anxiety, while 2.6% are affected by depression” could be rephrased as “Approximately 6.5% of children and adolescents are estimated to suffer from anxiety, while 2.6% are affected by depression (Polanczyk et al., 2015).”

Thank you for the suggestion. The suggested correction has been made (lines 47-48)

“Approximately 6.5% of children and adolescents are estimated to suffer from anxiety, while 2.6% are affected by depression (Polanczyk et al., 2015).”

Redundancy: The phrase “Studies primarily focus on the consequences observed in adulthood” is repeated multiple times throughout the manuscript. Consider consolidating these mentions for conciseness.

The sentence was changed to “and the focus is on the consequences observed in adult life” in Lines 88-89 and was deleted in Lines 159-160.

Methodology Section: Some sentences explaining the OWA operator and validation tests are lengthy and complex. Simplifying these sentences would enhance reader comprehension.

Thank you for the suggestion. The position of the text sections has been modified, and some adjustments have been made to the text to make the relationship between the proposed dimensions and the propensity for childhood anxiety and depression clearer. See the adjustments made to lines 127-231. The following excerpts were included:

[…] Constructing a composite indicator using the ordered weighted average operator is quite simple and involves the following steps:

Normalizing the sub-indicators.

Transposing the matrix of normalized sub-indicators \mathrm{\Omega}_1,\mathrm{\Omega}_2,\ldots,\mathrm{\Omega}_\rho.

Ordering the normalized sub-indicators in descending order.

Ordering the normalized sub-indicators in descending order.

Defining the weights \beta_1,\beta_2,\ldots,\beta_\rho according to the order matrix.

Calculate the composite indicator using the weighted average.

The normalization of sub-indicators performed in the first step can be performed using the following expression:

\mathrm{\Omega}_\lambda=\frac{\omega_\lambda-min\left(\omega_\lambda\right)}{max\left(\omega_\lambda\right)-min\left(\omega\right)} (1)

where \omega_\lambdarepresents the value of the sub-indicator λ for the decision unit ω, and max\left(\omega_\lambda\right) and min\left(\omega_\lambda\right) are the maximum and minimum values of the sub-indicator λ across all decision units ψ.

After transposing the normalized sub-indicator matrix in the second step, the sub-indicators are ordered from highest to lowest in the third step. In the fourth step, the sub-indicators are weighted. Note that the weights \beta depend on the intensity and direction of the emphasis. The choice of these parameters will define which lines of the matrix will be weighted with values different than zero.

For example, consider a multidimensional phenomenon with four sub-indicators. Then, assume that we want to emphasize the negative aspects of the multidimensional phenomenon. At this point, it is possible to establish a certain intensity of the negative emphasis. For example, it is possible to assign zero weight to the row of the matrix with the highest value and to emphasize the negative aspects slightly by assigning the weights \left(\beta=\frac{1}{\rho}\right) to the remaining rows that satisfy the conditions \beta_\rho\in\left[0,1\right] and \sum_{\lambda=1}^{\rho}\beta_\lambda=1.

Finally, the composite indicator is constructed in the fifth step using the following expression:

OWA\left(\mathrm{\Omega}_1,\mathrm{\Omega}_2,\ldots,\mathrm{\Omega}_\rho\right)=\sum_{\lambda=1}^{\rho}\beta_\lambda\alpha_\lambda (2)

where \alpha_\lambda corresponds to the λ-th highest value among \mathrm{\Omega}_1,\mathrm{\Omega}_2,\ldots,\mathrm{\Omega}_\rho, and the weights \beta_\lambda satisfy the conditions \beta_\lambda\in\left[0,1\right] and \sum_{\lambda=1}^{\rho}\beta_\lambda=1.

In short, after data normalization, the sub-indicators in descending order are weighted and then aggregated in a single score.

[…]

Terminology Consistency: Maintain consistent terminology throughout the manuscript. For instance, use “mental health conditions” or “anxiety and depression” uniformly unless contextually necessary to switch.

Thank you for the comment. In this case, these two terms are consistent with the context in which each appears in the text. The use of “mental health condition” is appropriate when referring to all mental health disorders to distinguish it from passages in which we refer only to issues of “anxiety and depression.” An adjustment has been made to the introduction to clarify this issue.

##################Reviewer #2: ##################

Thank you for the privilege of reading your article.

This article details the synthesis and validation of a composite indicator to assess and predict the risk of anxiety and depression in children and adolescents in different countries based on the findings of the multiple indicator cluster surveys collected by UNICEF. The opening of section 2 feels somewhat repetitive on reading and it is not entirely clear why it exists as a separate section from the introduction.

Thank you for the suggestion. To avoid repetition and make the text more consistent, adjustments have been made to the text and the order of the sections of the article. See lines 127-232.

In section 3, line 269 the sentence is incomplete - Research on adverse childhood experiences has predominantly focused … (on what?).

Thank you for the suggestion. The paragraph containing that sentence was excluded. Its content had already been adequately presented in the previous paragraph, but due to an editing error, it remained in the text.

The figure 7 needs to be labelled clearly.

Thank you for the suggestion. The text below was included to clarify:

“The distribution of the importance of each dimension in the composite indic

---

## [Decision Letter · Decision Letter 1]

28 May 2025

Propensity to Childhood Anxiety and Depression Due to Exposure to Adversity: A Multidimensional Construct

PONE-D-24-60103R1

Dear Dr. Mannan,

We’re pleased to inform you that your manuscript has been judged scientifically suitable for publication and will be formally accepted for publication once it meets all outstanding technical requirements.

Kind regards,

Javier Fagundo-Rivera, PhD

Academic Editor

PLOS ONE

Additional Editor Comments:

Recommendation: Accept

Dear Authors, you have thoroughly addressed all reviewer and editorial comments from the previous round, significantly improving the clarity, structure, and methodological transparency of the manuscript. The rationale for the selected dimensions is now well-justified, the OWA methodology is clearly explained, and the implications for public policy are thoughtfully articulated across different regional contexts. Minor issues related to terminology, figure labeling, and redundancy have also been resolved. While some technical sections may still be dense for general readers, they do not hinder the scientific integrity or clarity of the work. Overall, the manuscript is now suitable for publication in its current form.

Reviewers' comments:

Reviewer's Responses to Questions

**Comments to the Author**

1. If the authors have adequately addressed your comments raised in a previous round of review and you feel that this manuscript is now acceptable for publication, you may indicate that here to bypass the “Comments to the Author” section, enter your conflict of interest statement in the “Confidential to Editor” section, and submit your "Accept" recommendation.

Reviewer #1: All comments have been addressed

2. Is the manuscript technically sound, and do the data support the conclusions?

Reviewer #1: Yes

3. Has the statistical analysis been performed appropriately and rigorously? 

Reviewer #1: (No Response)

4. Have the authors made all data underlying the findings in their manuscript fully available?

Reviewer #1: Yes

5. Is the manuscript presented in an intelligible fashion and written in standard English?

Reviewer #1: Yes

6. Review Comments to the Author

Reviewer #1: The manuscript, "Propensity to Childhood Anxiety and Depression Due to Exposure to Adversity: A Multidimensional Construct," now presents a technically sound and methodologically rigorous piece of scientific research.

You have successfully addressed the comments raised in the previous review round. The justification for the inclusion of factors in each dimension (socioeconomic, vulnerability, and risk/insecurity) has been clarified and grounded in relevant literature. The expansion and clarification of the Ordered Weighted Averaging (OWA) methodology have significantly improved accessibility and transparency for a broad readership.

The statistical analysis has been conducted with appropriate rigor. The validation of the composite indicator through explanatory power, discriminant power, and outlier detection using Mahalanobis distance demonstrates a robust approach to ensure the reliability and interpretability of the findings.

The conclusions are well supported by the data and appropriately contextualized with relevant public policy recommendations for different regions. Furthermore, the study’s limitations, including reliance on secondary data and the non-causal nature of correlation, have been adequately acknowledged.The manuscript is well written in standard English and is presented in a clear, organized, and intelligible manner. Minor typographical issues raised earlier have been resolved.

Overall, this is an important and timely contribution to the literature on childhood mental health and the influence of adverse experiences. It provides valuable insights for researchers and policymakers aiming to mitigate the risk of anxiety and depression among children and adolescents globally.

7. PLOS authors have the option to publish the peer review history of their article (what does this mean? ). If published, this will include your full peer review and any attached files.

**Do you want your identity to be public for this peer review?** For information about this choice, including consent withdrawal, please see our Privacy Policy .

Reviewer #1: **Yes: ** Hala Awadd Ahmed

---

## [Editor Report · Acceptance letter]

PONE-D-24-60103R1

PLOS ONE

Dear Dr. Mannan,

I'm pleased to inform you that your manuscript has been deemed suitable for publication in PLOS ONE. Congratulations! Your manuscript is now being handed over to our production team.

Kind regards,

on behalf of

Dr. Javier Fagundo-Rivera

Academic Editor

PLOS ONE